# Barley Yellow Dwarf Virus Influences Its Vector’s Endosymbionts but Not Its Thermotolerance

**DOI:** 10.3390/microorganisms12010010

**Published:** 2023-12-19

**Authors:** Evatt Chirgwin, Qiong Yang, Paul A. Umina, Joshua A. Thia, Alex Gill, Wei Song, Xinyue Gu, Perran A. Ross, Shu-Jun Wei, Ary A. Hoffmann

**Affiliations:** 1Cesar Australia, 95 Albert Street, Brunswick, VIC 3056, Australia; pumina@unimelb.edu.au; 2PEARG Group, School of BioSciences, Bio21 Institute, The University of Melbourne, Parkville, VIC 2052, Australia; joshua.thia@unimelb.edu.au (J.A.T.); alex.gill@unimelb.edu.au (A.G.); xinyue.gu1@unimelb.edu.au (X.G.); perran.ross@unimelb.edu.au (P.A.R.); ary@unimelb.edu.au (A.A.H.); 3Institute of Plant Protection, Beijing Academy of Agriculture and Forestry Sciences, Beijing 100097, China; songw0513@163.com (W.S.); shujun268@163.com (S.-J.W.)

**Keywords:** oat aphids, pest, agriculture, biocontrol, thermotolerance, luteovirus, insects, GxG, climate, symbionts

## Abstract

The barley yellow dwarf virus (BYDV) of cereals is thought to substantially increase the high-temperature tolerance of its aphid vector, *Rhopalosiphum padi*, which may enhance its transmission efficiency. This is based on experiments with North American strains of BYDV and *R. padi*. Here, we independently test these by measuring the temperature tolerance, via Critical Thermal Maximum (CTmax) and knockdown time, of Australian *R. padi* infected with a local BYDV isolate. We further consider the interaction between BYDV transmission, the primary endosymbiont of *R. padi* (*Buchnera aphidicola*), and a transinfected secondary endosymbiont (*Rickettsiella viridis)* which reduces the thermotolerance of other aphid species. We failed to find an increase in tolerance to high temperatures in BYDV-infected aphids or an impact of *Rickettsiella* on thermotolerance. However, BYDV interacted with *R. padi* endosymbionts in unexpected ways, suppressing the density of *Buchnera* and *Rickettsiella*. BYDV density was also fourfold higher in *Rickettsiella*-infected aphids. Our findings indicate that BYDV does not necessarily increase the temperature tolerance of the aphid transmission vector to increase its transmission potential, at least for the genotype combinations tested here. The interactions between BYDV and *Rickettsiella* suggest new ways in which aphid endosymbionts may influence how BYDV spreads, which needs further testing in a field context.

## 1. Introduction

Barley and cereal yellow dwarf viruses (henceforth, BYDV) encompass the most damaging viruses to cereal crops worldwide [1,2,3]. However, BYDV requires biological vectors to infect new plants with aphids being their primary vector [1,4]. Consequently, the risk that BYDV poses to crops is intertwined with the ecology and transmission efficiency of their vectors. The success of the most common BYDV serotype worldwide, BYDV-PAV, has been facilitated by the wide distribution and efficient transmission of its primary vector, the bird-cherry oat aphid—*Rhopalosiphum padi*, Linnaeus (Hemiptera: Aphididae) [1,5]. Agricultural management strategies for BYDV-PAV have largely relied on the insecticide control of *R. padi* [2,6]. However, the emergence of insecticide resistance in *R. padi* and other BYDV vectors means alternative methods of disrupting the relationship between *R. padi* and BYDV-PAV are needed [7,8,9,10].

Viruses often improve transmission efficiency by altering their vector’s phenotype [11,12,13], and growing evidence suggests BYDV-PAV alters *R. padi* in multiple ways [14,15]. Recently, Porras et al. [16] discovered that BYDV-PAV substantially (8 °C) enhanced the Critical Thermal Maximum (CT_max_) of viruliferous *R. padi* by triggering aphids to upregulate heat shock proteins. BYDV-PAV was also found to increase the surface temperature of the infected host plants (wheat, *Triticum aestivum* L.). In doing so, viruliferous *R. padi* may gain an advantage over other aphid species feeding on an infected plant. In their study, Porras et al. [16] tested a single North American strain of BYDV-PAV and a single *R. padi* colony, and it remains unclear if this temperature-based relationship generalizes to other isolates and strains around the world where BYDV-PAV and *R. padi* are economically damaging. This is important to establish from an economic perspective given the rate of spread of BYDV-PAV can be affected by warm conditions. For instance, warmer conditions may alter the titer of BYDV-PAV in infected plants, and warmer conditions may also shorten the latent period (the time between a vector first acquiring and transmitting the virus) of *R. padi* carrying BYDV [17,18].

Endosymbionts offer new avenues to manage agricultural pests and vector-transmitted plant viruses [19,20,21,22]. Endosymbionts include the heritable bacteria, fungi, and/or viruses hosted within aphids and many other taxonomic groups, which can alter their host’s phenotypes and ecology [23,24,25]. Amongst aphids, endosymbionts are commonly categorized as being primary or secondary [24]. *Buchnera aphidicola* (Enterobacterales: Erwiniaceae; henceforth referred to by genus) is the sole primary endosymbiont for most aphid species and provides the essential amino acids that aphids require for survival and reproduction [26]. *Buchnera* is thought to play a critical role in the aphid transmission of BYDV, although the exact mechanisms involved are debated [27,28,29,30]. Secondary endosymbionts are unnecessary for host survival but can improve their host’s fitness in some contexts, such as providing protection against predators or pathogens [31,32,33]. In some cases, secondary endosymbionts (e.g., *Wolbachia*) can disrupt their host’s ability to transmit viruses [22,34], although others (e.g., *Rickettsia*) can enhance virus transmission [35,36]. BYDV transmission by *Sitobion miscanthi* appears to be enhanced by *Rickettsia* [37], but so far, no endosymbiont has been shown to disrupt BYDV transmission.

The secondary endosymbiont *Rickettsiella viridis* (*Legionellales*: *Coxiellaceae*; henceforth referred to by genus) may offer a novel pathway for reducing the thermal benefits that *R. padi* gains from BYDV. *Rickettsiella* naturally occurs in pea aphids (*Acyrthosiphon pisum*), where it protects its host against fungal pathogens [38]. Recently, Gu et al. [39] artificially introduced *Rickettsiella* to the green peach aphid (*Myzus persicae)* and discovered that this transinfection could spread in laboratory-based populations via plant-mediated horizontal transmission and vertical transmission. *Rickettsiella* infection also altered multiple *M. persicae* phenotypes (e.g., fecundity), including a reduction in their CT_max_ and heat knockdown time [39]. If *Rickettsiella* has a similar effect on *R. padi,* then this endosymbiont may offer a novel tool for disrupting a temperature-dependent relationship between *R. padi* and BYDV.

Here, we investigate the relationship between *R. padi,* BYDV and temperature tolerance using Australian specimens, and we also consider how *Rickettsiella* infection affects these interactions. Specifically, our study set out to address three questions. (1) Does an Australian strain of BYDV-PAV provide *R. padi* the same enhanced thermotolerances as reported in the North American strain? (2) Does the introduction of *Rickettsiella* change the thermotolerance of viruliferous and non-viruliferous *R. padi*? (3) Do *Buchnera*, *Rickettsiella* and BYDV-PAV alter the densities of each other? To do so, we created factorial combinations of aphids infected with BYDV-PAV and *Rickettsiella* and then measured their thermotolerance in multiple ways. We also sampled individual aphids unexposed to heat treatments and measured BYDV-PAV, *Rickettsiella* and *Buchnera* densities to explore the interactions between these microbes.

## 2. Materials and Methods

### 2.1. General Outline of Experimental Design

We completed our experiment below over multiple blocks due to the logistical constraints of simultaneously culturing and assaying the required number of aphids and plants. In each block, we grew a separate cohort of plants that were then used to create a cohort of viruliferous and non-viruliferous aphid lines to investigate our three study questions. Aphid thermotolerance (Section 2.4) and the interactions between BYDV and aphid endosymbiont densities (Section 2.5) were initially measured together over two blocks. Following these first two blocks, our results for aphid thermotolerance across all four groups were reasonably clear, but the influence of *Rickettsiella* infection on the BYDV density of aphids remained more equivocal. Therefore, we completed a third block that solely tested the interactions between BYDV and aphid endosymbiont densities to clarify the relationship between *Rickettsiella* infection and the BYDV density of viruliferous aphids.

### 2.2. Maintenance of Virus Isolate

The isolate of BYDV-PAV was kindly provided by Dr. Piotr Trebicki (Grains Innovation Park, Horsham, VIC, Australia) and maintained at low density on *T. aestivum* c.v. Trojan. PCR amplicons of the coat protein gene (600 bp) of BYDV were sequenced in both forward (BYL, Table 1) and reverse (BYR, Table 1) directions using Sanger Sequencing (Macrogen, Inc., Geumcheongu, Seoul, Republic of Korea). The sequences were analyzed with Geneious 9.18 software. A phylogenetic tree was constructed with MEGA. Throughout our study, the Trojan wheat variety was used as the host plant for *R. padi* and BYDV. Wheat seedlings were grown in a 22 °C Controlled Temperature (CT) room with a 14:10 (L:D) h photoperiod.

### 2.3. Aphid Line Creation and Maintenance

The Grains Innovation Park (Horsham, VIC, Australia) provided the isofemale line of *R. padi* used in our study, and this line was maintained in the laboratory asexually on *T. aestivum* leaves placed in 10 g/L agar in Petri dishes at 12 °C with a 14:10 (L:D) h photoperiod for >40 generations (~3 years) prior to being used in this study. To test our study questions, we used the four factorial combinations of *Rickettsiella* and BYDV: *Rickettsiella* positive and viruliferous (R+V+), *Rickettsiella* negative and viruliferous (R−V+), *Rickettsiella* positive and non-viruliferous (R+V−), and *Rickettsiella* negative and non-viruliferous (R−V−). As already noted above, we completed our experiment with these lines over three blocks due to logistical constraints.

The R+ line used in this study was created by introducing *Rickettsiella* into our *R. padi* line from *A. pisum,* which was originally collected from lucerne (*Medicago sativa* L). *Rickettsiella* was transferred using microinjection [43], whereby the hemolymph from donor aphids (*A. pisum*) was transferred to *R. padi*, and one of the surviving *R. padi* infected with *Rickettsiella* was used to establish the R+ isofemale line on wheat. *Rickettsiella* has now stably infected its host over 30 aphid generations. 

Producing the viruliferous (V+) and non-viruliferous (V−) aphids for each block required three steps. Unlike endosymbionts, aphids do not directly pass BYDV to their offspring, and each generation must feed on BYDV-infected (BYDV+) plants to become inoculated with the virus. Accordingly, we created V+ and V− aphid lines via three steps: (1) infecting plants with BYDV; (2) culturing a sufficient number of R+ and R− aphids at the same age, and (3) inoculating the age-matched aphids (or leaving them un-inoculated for V− aphids) by placing them on BYDV+ plants to feed. Full details of each step are provided below. Aphids were maintained at 20 °C in a CT room with a 14:10 (L:D) h photoperiod using 2400 Lumen lights for all three steps. Further details of each of the three steps are provided below.

For each block, we grew ~30 *T. aestivum* plants to the 3-leaf stage (~3 weeks) in soil (Osmocote potting mix) and housed them within an insect-proof mesh container (93 × 47.5 × 47.5 cm). At the 3 leaf-stage, half the plants cultured in each block were inoculated with BYDV (used to create V+ aphid lines), and the other half was left uninoculated (to create V− aphid lines). Each plant was inoculated by placing the tip of its second true leaf into a 55 mL vial containing ten viruliferous aphids and then sealing this vial with cotton wool. After one week of inoculation, all aphids were removed from plants, and the plants were left for 14 days to allow the virus to spread. We concurrently completed the same steps for each uninoculated plant using non-viruliferous aphids to ensure that the V− plants remained valid controls. After 14 days, we screened plants for BYDV infection, and any plants in the V+ group that had failed to become infected through viruliferous aphids were discarded.

Next, we ensured all aphids tested in our study were the same age and life stage by setting up ten age-matching plates for each of the R− and R+ lines. Each age-matching plate consisted of 30 adult aphids within a 100 mm petri dish containing *T. aestivum* leaves placed in 10 g/L agar. The age-matching plates produced ~500 R− or R+ 1–2-day-old nymphs. Half of these nymphs were placed on BYDV+ plants for three days to become inoculated (creating the R+V+ and R−V+ lines), and the other half remained in control BYDV- plants (creating R+V− and R−V−) lines. We selected a three-day inoculation period based on previous pilot studies, which showed virus inoculation in >99% of aphids during this time. Therefore, all phenotypic and endosymbiont measurements below were from 4–5-day-old aphids.

### 2.4. Measuring Thermotolerance

We measured thermotolerance in two ways: CT_max_ and heat knockdown time. Individual aphids were placed in glass tubes on a rack within a programmable water bath (Ratek Thermoregulator—Digital Immersion Heater Circulator). The water bath began at 22 °C for 10 min to allow aphids to acclimatize with temperature then increasing by 0.2 °C per min to 35 °C and 0.1 °C per min from 35 °C until all aphids were incapacitated (unable to self-right). To determine CT_max_, each aphid was visually inspected to find the temperature when they ceased moving or could not right themselves. Heat knockdown time was measured by exposing aphids to a constant temperature (40 °C) and recording the time aphids ceased moving or could not right themselves. A temperature of 40 °C was used based on previous pilot studies, which showed this to be the average CT_max_ of R−V− *R. padi*. All experiments were run blindly with respect to aphid line. Lines were created to measure CT_max_ and knockdown time over two separate experimental blocks. Given a limited number of aphids could be scored (i.e., visually inspected) simultaneously in each thermotolerance assay, we measured each thermotolerance trait over multiple runs (i.e., heat exposure events) with the aphids from each block. Overall, the two experimental blocks included the four experimental runs of each thermotolerance trait, whereby we measured the CT_max_ of 231 aphids (≥57 from each group) and the knockdown time of 187 aphids (≥45 from each group).

### 2.5. Measuring Endosymbiont and BYDV Density

To explore the interaction between aphid endosymbionts and BYDV, we tested the *Buchnera*, *Rickettsiella*, and BYDV density of aphids that were not exposed to a heat treatment. Over the three blocks, we measured the endosymbiont density of 210 aphids (≥35 from each group) and the virus density of 76 aphids (≥37 from each of the V+ groups).

Our first step in screening endosymbiont and BYDV density was to extract the total RNA from individual aphid samples using a Monarch Total RNA Miniprep Kit (NEB, Ipswich, MA, USA). First, 300 ng RNA from each sample was reverse transcribed into cDNA using a high-capacity cDNA reverse transcription kit (Thermo Fisher, Waltham, MA, USA), which was then used as the template for qPCR assays with a Roche LightCycler 480 using a High-Resolution Melting Master kit (Roche Diagnostics Australia Pty. Ltd., North Ryde, NSW, Australia) and IMMOLASE^TM^ DNA polymerase (5 U/µL) (Bioline AgroSciences, Camarillo, CA, USA) according to [41].

Four primer sets (Table 1) were used to amplify markers specific to BYDV-PAV, *Buchnera*, *Rickettsiella* and *R. padi* β-actin. Two–three consistent replicate runs were averaged, and these average values were subsequently used in the data analysis (Section 2.6) below. Delta crossing point (Cp) values were calculated by subtracting the Cp value of the BYDV, *Buchnera* and *Rickettsiella*-specific marker from the Cp value of the β-actin marker. The standard deviation (SD) was calculated with delta Cp value of the 2–3 technical replicates. The replicates were considered valid when the SD was <1. Delta Cp values of valid replicates were transformed by 2^n^ to produce relative endosymbiont or BYDV density measures.

### 2.6. Data Analysis

We ran a series of linear mixed-effects models to test our three study questions using the ‘glmmTMB’ package in R version 4.0.2 [44]. This modeling approach allowed us to test whether our traits of interest (e.g., CT_max_) differed between treatment groups (i.e., fixed-effect predictors) whilst also using random-effects predictors to account for aphids being tested from the same block or run. Diagnostics checks of the assumptions of linear models (i.e., normality and variance amongst treatment groups) were assessed using the ‘DHARMa’ package [45,46]. The raw values for thermotolerance traits (CT_max_ and knockdown time) were used in our models (detailed below) because these traits showed an approximately normal distribution. However, the BYDV and endosymbiont density values were log-transformed to normalize their distribution when included in the models below. We employed Wald chi-square (χ^2^) tests to assess the significance of fixed-effects predictors in all models below using the ‘car’ package [47].

First, we tested whether *Rickettsiella* and BYDV influenced the CT_max_ or knockdown time of *R. padi* in two separate linear mixed-effects models. In each model, thermotolerance (CT_max_ or knockdown time) was the response variable with *Rickettsiella* status, BYDV status, and their interaction as categorical fixed-effect predictors. Each run was treated as a nested random effect within the blocks in both models to account for aphids being tested at the same time and/or originating from the same cohort of aphids and plants.

Second, we tested whether endosymbiont density responded to BYDV status using two separate linear mixed-effects models. The *Buchnera* density of aphids was modeled as a response variable with *Rickettsiella* status, BYDV status, and their interaction as fixed-effects predictors and block as a random-effect predictor. Next, the *Rickettsiella* density of aphids was modeled as a response variable to BYDV status as a single fixed-effect predictor and block as a random-effect predictor.

Third, we explored whether the *Rickettsiella* status of *R. padi* affected the BYDV density carried by aphids. To do this, we ran a linear mixed-effects model that included *Rickettsiella* status as a fixed-effect predictor and block as a random-effect predictor.

Finally, we tested whether the BYDV density of individual aphids significantly covaried with *Buchnera* and/or *Rickettsiella* density. To do this, BYDV density was used as the response variable in a linear mixed-effects model with *Buchnera* density, *Rickettsiella* density, and their interaction as fixed-effect predictors and block as a random-effect predictor.

### 2.7. Comparison of Genomic Backgrounds

We compared the genomic background of our focal *R. padi* strain (OAT_02) against strains sampled in the Australian states of Victoria (*n* = 7), New South Wales (*n* = 2), and South Australia (*n* = 2) as well as the North American strain from Porras et al. [16]. We estimated pairwise differentiation across all sample pairs with Δ_D_ statistics [48], which estimates the proportional differentiation among samples. Full details on the genomic comparison methods are provided in the Appendix A.

## 3. Results

Thermotolerance traits and *Rickettsiella* density varied across experimental blocks and/or runs, but *Buchnera* and BYDV densities were consistent across blocks. All block and run effects were accounted for by including them as random predictors in the models below.

### 3.1. Thermotolerance

Neither BYDV nor *Rickettsiella* infection significantly altered the thermal tolerance of *R. padi* (Figure 1). On average, CT_max_ was 39.9 °C, which was not altered by BYDV (χ^2^ = 1.64, d.f. = 1, *p* = 0.20) or *Rickettsiella* infection (χ^2^ = 1.55, d.f. = 1, *p* = 0.21) nor was there an interaction between BYDV and *Rickettsiella* infection status (χ^2^ = 0.55, d.f. = 1, *p* = 0.46). Similarly, knockdown time was unaffected by BYDV (χ^2^ = 0.41, d.f. = 1, *p* = 0.52) and *Rickettsiella* infection (χ^2^ = 0.99, d.f. = 1, *p* = 0.32) nor was there an interaction between them (χ^2^ = 0.004, d.f. = 1, *p* = 0.99).

### 3.2. BYVD and Endosymbiont Interactions

*Buchnera* (Figure 2a; χ^2^ = 14.60, d.f. = 1, *p* < 0.01) and *Rickettsiella* (Figure 2b; χ^2^ = 7.80, d.f. = 1, *p* = 0.01) densities were lower in V+ aphids than V− aphids, although there was considerable overlap between treatment groups. On average, V− aphids had approximately double the *Buchnera* density and five times the *Rickettsiella* density of V+ aphids. *Buchnera* density was not significantly affected by *Rickettsiella* infection (χ^2^ = 0.67, d.f. = 1, *p* = 0.41) nor was there an interaction between BYDV and *Rickettsiella* infection (χ^2^ = 2.22 d.f.= 1, *p* = 0.14).

On average, BYVD density was four times higher in aphids infected with *Rickettsiella* than in aphids uninfected with *Rickettsiella* (Figure 3: χ^2^ = 8.09 d.f. = 1, *p* < 0.01). However, across individual *R. padi*, there was no significant covariance between the relative density of BYVD and *Rickettsiella* (Appendix A: χ^2^ = 0.99, d.f. = 1, *p* = 0.32) or *Buchnera* (Appendix A: χ^2^ = 0.27, d.f. = 1, *p* = 0.61) density.

### 3.3. Comparison of Genomic Backgrounds

For BYDV-PAV, the phylogenetic tree suggested our BYDV-PAV isolate is quite distant from all the other isolates published in GenBank, including Australian and New Zealand isolates (Appendix A). 

After trimming, our samples of Australian *R. padi* had an average of 41,590,394 reads with a range of 39,325,647 and 43,839,140 reads. The North American *R. padi* had 29,617,835 reads after trimming. After all SNP filtering steps, we were left with 6694 SNPs. Note that because we filtered for no missing data, all SNPs come from protein-coding regions of the genome (due to the use of the North American strain transcriptome). Our results show that all Australian *R. padi* sampled in this study had a very similar genomic background. Pairwise Δ_D_ was <0.002 for all Australian pairs. The North American strain was quite different to all Australian clones with pairwise Δ_D_~0.19 in all comparisons (Appendix A). This suggests that our study used a different genomic background for *R. padi*.

## 4. Discussion

An understanding of BYDV’s relationship with their aphid vectors and disruptive influences may provide new avenues to manage this virus. Recently, Porras et al. [16] showed that viruliferous *R. padi* gained enhanced thermotolerance (CT_max_ up by 8 °C) that may provide a competitive advantage over other aphids under warm conditions. Here, we explored the relationship between *R. padi*, BYDV and temperature tolerance using Australian aphid and BYDV material and examined whether this temperature-based relationship could be disrupted by the endosymbiont *Rickettsiella*. We found that neither BYDV-PAV nor *Rickettsiella* significantly altered *R. padi* thermotolerance (CT_max_ and knockdown time). As such, our findings suggest the Australian strains of BYDV-PAV and *R. padi* tested here interact differently to the North American strains of Porras et al. [16]. However, somewhat unexpectedly, *Rickettsiella* infections appeared to increase the BYDV density carried by *R. padi*, which may influence BYDV transmission.

Genetic differences between the North American and Australian strains of BYDV-PAV, *R. padi*, and *T. aestivum* or their interactions may explain why our results differed from those of Porras et al. [16]. Our analyses of genomic backgrounds suggest that considerable genetic differentiation exists between the *R. padi* and BYDV-PAV used in the two studies (Appendix A). Some of this genetic differentiation may reflect adaptation in BYDV and/or *R. padi* to different climates. Indeed, the non-viruliferous *R. padi* here showed a much higher CT_max_ (5 °C) than those in Porras et al. [16]. However, testing conditions could also account for the different CT_max_ values between the two studies [49]. Alternatively, genetic interactions (G × G) often shape the performance of vectors, hosts and pathogens [50,51], and these are possible between all three of the biological levels examined here (BYDV, *R. padi*, or *T. aestivum*). G × G interactions are well established in other insect–pathogen systems [52,53], but they have been rarely tested between BYDV and *R. padi* (but see [54]). We also note that our focal *R. padi* strain had a similar genomic background to the 13 other Australian strains included in our genomic comparisons. It therefore seems unlikely that we would have observed a different result by using any of the other *R. padi* strains available to us. Subsequent work to identify different genomic backgrounds of Australian *R. padi* is underway, which will facilitate the future testing of G × G interactions.

Plasticity may also contribute to the differences between our findings and those of Porras et al. [16]. *Rhopalosiphum padi* has repeatedly been shown to increase their thermotolerance via acclimation [55,56]. Still, the increase in thermotolerance due to BYDV noted by Porras et al. [16] was extremely large and unlikely to be explained by acclimation alone. Therefore, we were surprised that there was no evidence of any effect in our data despite similarities in the assays used. In both studies, aphids were maintained at the same temperature (20 °C) for multiple generations preceding exposure; CT_max_ assays ramped temperature at a similar rate (0.2–0.1 °C per min) and tested aphids at a similar developmental stage (~4 days old). Nonetheless, the two studies differed in some ways (e.g., exposure via hotplate versus water bath), and other lab-based conditions (e.g., soil, watering volume, and the quality of the growth chambers) are likely to have differed, which could have influenced the results. Even small methodological differences in CT_max_ assays can obscure biological patterns in thermotolerance [57,58], and conjecture remains on the best methods to measure CT_max_ [59].

There are contrasting hypotheses as to why BYDV suppressed both endosymbionts of *R. padi*. First, endosymbiont suppression (particularly primary endosymbionts) is often associated with stressful conditions (e.g., temperature and chemicals) [41,60,61], and the presence of BYDV may be stressful for *R. padi* and/or its endosymbionts. The suppression of *Buchnera* density may further decrease the capacity of *R. padi* to synthesize essential vitamins, which can cause severe fitness costs [26,62]. Second, lower endosymbiont densities in *R. padi* could be the result of BYDV improving the nutritional content of *T. aestivum* [36,63]. For example, the potato leafroll virus causes host plants to produce more essential amino acids (e.g., argE) that their insect vectors otherwise must obtain from their nutritional endosymbionts [63]. Hence, lower endosymbiont densities may reflect insect vectors becoming less reliant on endosymbionts for nutrition (assuming that endosymbiont density is under host control) [64]. Wheat plants infected with BYDV have a higher essential amino acid content [15], meeting one of the requirements of this hypothesis. The two hypotheses could be tested further using membrane feeders to infect sucking insects with a virus without changing their diet [65,66,67].

Whether the increased virus density carried by *R. padi* hosting *Rickettsiella* will translate to higher rates of BYDV transmission warrants further exploration. The relationship between virus density and transmission rate for persistent viruses like BYDV (as opposed to viruses that are only intermittently carried inside vectors) is unclear [68]. For example, Rotenberg et al. [69] found Western flower thrips (*Frankliniella occidentalis*) carrying higher titers of a persistent virus (Tomato spotted wilt virus) transmitted this virus with a higher frequency. Conversely, other studies have found that virus density is less important compared with other factors like virus isolate for a vector’s transmission of persistent plant viruses [68,70]. The effect that *Rickettsiella* has on BYDV transmission could have important implications for deploying this endosymbiont for pest-control purposes (cf. [39]) and highlights the important interactions between viruses and endosymbionts more generally.

Further work could consider testing multiple isolates of BYDV and multiple *R. padi* clones. BYDV-PAV isolates have diverged in their genetics and pathogenicity as the virus has spread across the world [71], which makes generalizing any results from a single isolate a challenge. In addition, we only tested a single clonal type, but multiple *R. padi* clones are present in populations [72,73] and can differ substantially in terms of life history characteristics [74]. Other measures of thermotolerance beyond CT_max_ and knockdown time could be considered given that the aphid life stage when heat exposure occurs can impact thermotolerance [75,76,77]. Sublethal life history traits are also influenced by stage-specific exposures to heat stress [78,79], which could shape the interactions between BYDV and *R. padi* under field conditions.

## 5. Conclusions

Our study suggests, at least for the Australian genotype combinations tested here, that BYDV does not necessarily enhance the temperature tolerance of *R. padi*. As such, our findings point toward key regional variations in the relationship between these two widespread pests. The interactions between BYDV and *Rickettsiella* indicate new ways aphid endosymbionts may influence how BYDV spreads. Still, further testing, involving multiple genotypes and in a field context, is needed to determine whether future BYDV management strategies can exploit this endosymbiont interaction.

## Figures and Tables

**Figure 1 microorganisms-12-00010-f001:**
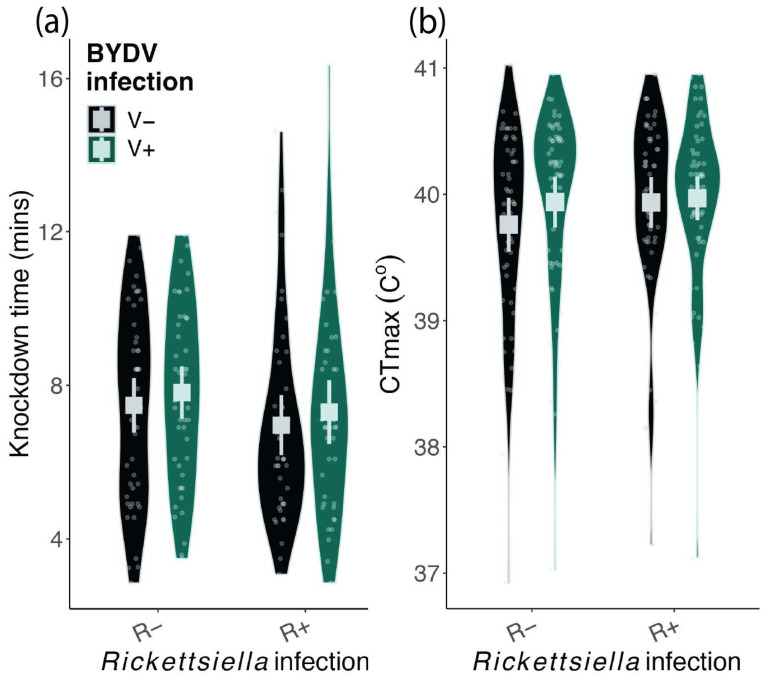
Knockdown time (**a**) and CT_max_ (**b**) of viruliferous (V+) and non-viruliferous (V−) *Rhopalosiphum padi* carrying (R+) or lacking (R−) *Rickettsiella viridis*. Squares show mean values and error bars represent 95% Confidence Intervals. Violin plots visualize the distribution of the data and the density of values from individual aphids.

**Figure 2 microorganisms-12-00010-f002:**
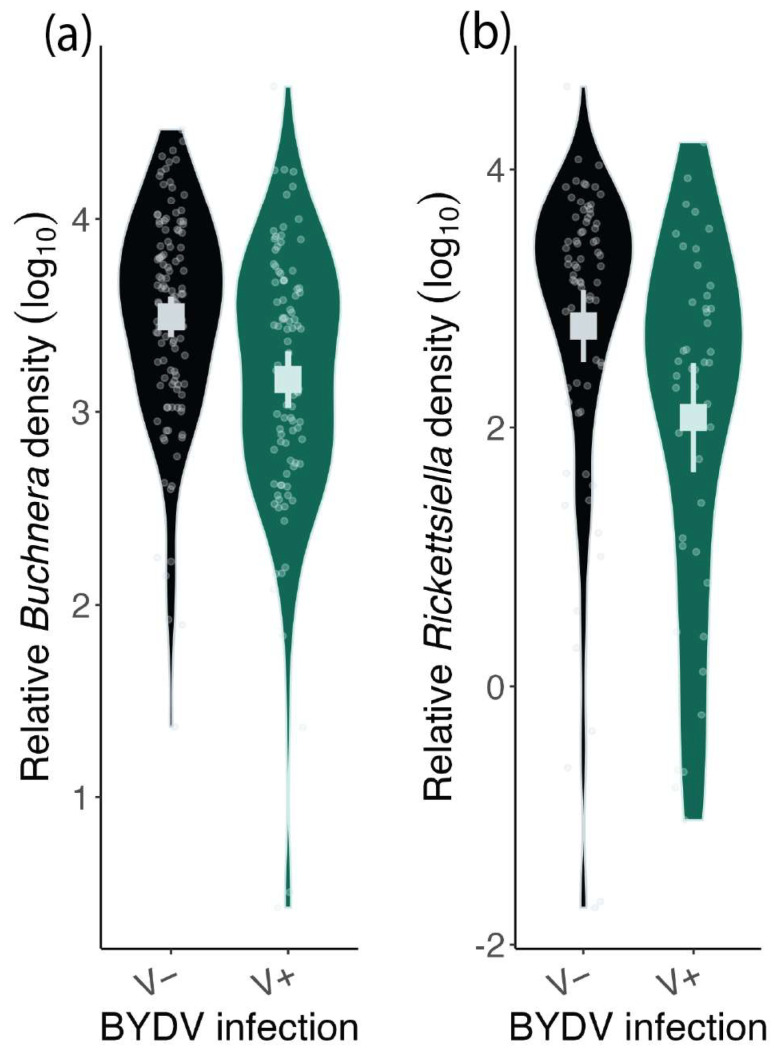
The relative *Buchnera aphidicola* (**a**) and *Rickettsiella viridis* (**b**) density of viruliferous (V+) and non-viruliferous (V−) *Rhopalosiphum padi*. *Buchnera* densities are pooled across R+ and R− aphids as *Rickettsiella* status did not impact *Buchnera* density. Squares show mean values, and error bars represent 95% Confidence Intervals. Violin plots visualize the distribution of the data and the density of values of individual aphids. Note: the relative densities are plotted on a log scale.

**Figure 3 microorganisms-12-00010-f003:**
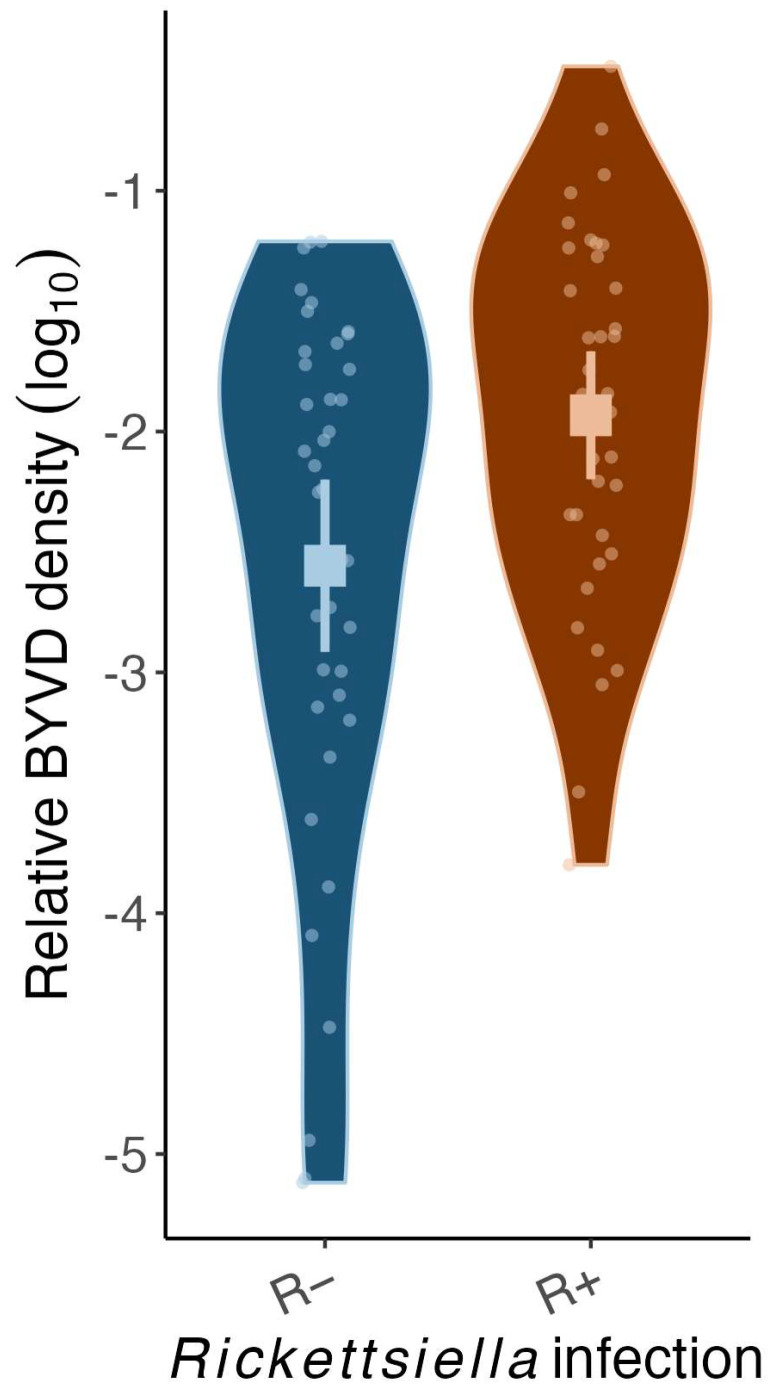
The relative BYDV density of *Rhopalosiphum padi* carrying (R+) or lacking (R−) *Rickettsiella viridis*. Squares show mean values and error bars represent 95% Confidence Intervals. Violin plots visualize the distribution of the data and the density of values of individual aphids. Note: the relative density is plotted on a log scale.

**Table 1 microorganisms-12-00010-t001:** Primers used for the qPCR-based detection of BYDV and endosymbionts in *Rhopalosiphum padi*.

Organism Targeted	Primer Name	Primer Sequence	Reference
BYDV-PAV	BYL	GTGAATGAATTCAGTAGGCCGT	[40]
	BYR	GTTCCGGTGTTGAGGAGTCT	
*Buchnera*	Buch_16S_F1c	AAAGCTTGCTTTCTTGTCG	[41]
	Buch_16S_R1a	GGGTTCATCCAAAAGCATG	
*Rickettsiella*	RCL16S-211F	GGGCCTTGCGCTCTAGGT	[42]
	RCL16S-470R	TGGGTACCGTCACAGTAATCGA	
β-actin	actin_aphid_F1	GTGATGGTGTATCTCACACTGTC	[41]
	actin_aphid_R1	AGCAGTGGTGGTGAAACTG	

## Data Availability

Our raw Illumina sequence reads have been deposited into Figshare: https://figshare.com/articles/dataset/Oat_aphid_i_Rhopalosiphum_padi_i_clone_bioinformatics/24540361 (accessed on 10 November 2023).

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
