# Peer review of "Barley Yellow Dwarf Virus Influences Its Vector’s Endosymbionts but Not Its Thermotolerance"

_microorganisms, 2023, doi:10.3390/microorganisms12010010_

Round 1

Reviewer 1 Report

Comments and Suggestions for Authors

Barley yellow dwarf virus is a global problem in cereal production. A better understanding of how it interacts with the vector is a key element in understanding the biology of the virus in agroecosystems. To that end, it is important to understand how the virus interacts with the vector and other organisms therein.

I liked this paper because it is partly a replicated study of earlier work by independent researchers. This is a key element in the scientific process and must be published. 

The supplemental material was uninterpretable because the EndNote citations were left as codes rather than citations. I could copy-paste some content to view without codes, but it did not always work. The figure with regression did not provide metrics showing model fit, p-value, R2, or appropriateness regarding model assumptions. It may be there, but I cannot tell.

Line 14) CTmax is a shortened form of what? I could see something like CT50 which is the critical temperature at which 50% of the aphids cannot move due to heat stress. Line 112, CT is controlled temperature, so CTmax is the maximum controlled temperature where none of the aphids are responding? Maybe CTmax is just over the maximum temperature each aphid was able to survive.

Line 38) The text after “(Hemiptera: Aphididae).” is not a sentence.

Line 50) The virus cannot do this at the same time.

Line 57) Reminds me of degree day models. Please clarify “shortens the latent period.” Under warmer temperatures it takes less time to accumulate a specific number of thermal units. However, it is also possible that the latent period becomes shorter when accumulated thermal units are accounted for. Is the perceived “shorter latent period” simply due to more rapid accumulation of thermal units or does it also require fewer thermal units?

Line 110) How does this fit within the limits of the subsection?

Line 117) I have missed where you describe rearing conditions in the laboratory: temperature, photoperiod.

Line 139) Were the 30 plants in individual containers, or in flats?

Line 139) Soil or hydroponic? Fertilized, or plants were used before any was necessary to maintain good plant growth?

Line 139) How is this a part of “Aphid lines and maintenance?” You need to change subsection titles to be more representative of contents or you need to add more subsections.

Line 139) I assume you are defining “block.” So a block is 30 plants, 15 will be BYVD free and the other 15 are BYVD infected.

Line 151) I think you mean or rather than and.

Line 151) As written this does not happen. Did you happen to remove the adults at some point? If not, then the nymphs are 0-2 days old if you started the experiment 48 H after adding adults to a plate.

Line 165) One interpretation: You completed the same steps for the uninoculated plants, yet the last step for the inoculated plants was to discard any uninfected plants. Presumably that means that all the uninfected plants were discarded and there was nothing left?

Line 169) “Lines were created” so these are different that the lines created earlier?

Line 172) It would not make sense if blocks as used here are the same as blocks used in line 139. What is a block? Four runs across two blocks? How are groups distributed within blocks?

Suggestion: Start the methods with an outline of the experiments. Just enough that I know what the treatments are, can follow the analysis, and have some idea of replication. If “block” refers to a “blocking variable” in a statistical model, then clearly define block in each section of this research.

Line 176) How were they tested?

Line 181) Technically this is a bad design. The key term is p-hacking. You ran the experiment and did not get conclusive results. You ran another replicate (block?), got significant results, and stopped.

Line 184) Do you means samples consisting of individual aphids or individual samples consisting of multiple (how many?) aphids?

Line 190) It appears that you are mixing testing for BYVD into a section on endosymbionts.

Line 194) What do you mean by replicate runs? If a replicate run means 2 to 3 PCR plate wells from each sample, these are better handled by averaging them rather than including a “run” variable in the analysis. Having this sort of replicate run is essential to minimize laboratory error effects, but you do not explore this source of variation further. Including run in the model makes models needlessly complex and does not further our understanding of the biological system.

Line 195) The analysis is not explained very well. Consider that many readers will not know of this method and will need help. This is the first time I have encountered the glmmTMB package. I look in Brooks et al. 2017 and I cannot translate their examples into your outcomes. For example, there are no obvious Chi-square tests in Brooks et al. 2017, but they are the first thing encountered in your results. Your analysis may be the best one and you may have used the best package for this analysis, but you have not made that case.

     The first model I think of is a probit model typically used to model insecticide dose-response data. At least superficially the CTmax data would appear to be appropriate for this type of analysis. On the other hand, with Figure 1, I would have a hard time seeing any conclusion other than the one presented.

Figure 1) In the methods you cite Brooks et al 2017. In the introduction Brooks states “Observed response variables are often in the form of discrete count data.” I am uncertain how a “Knockdown time (mins)” fits the zero inflated Poisson model. At least with CTmax I can sort of see the data as count data where at each interval I ask every aphid “are you dead?” and each point is where an individual succumbed to thermal stress.

Figure 2) Is this density per aphid, or density per microliter aphid extract, or something else?

Figure 2) How accurate is the quantification method with estimated densities less than ten?

Figure 2) An estimate of less than 1 Rickettsiella seems dubious. I see how you get the value, but I fail to see how the PCR method is reliable with this outcome.

Figure 2) At least I see this as count data. The zero-inflated part is elusive as all aphids had the endosymbionts.

Line 301) Say (for argument) that the response observed by Porras et al 2020 was entirely due to the heat shock proteins identified in that paper. How sensitive is your genetic testing to those specific proteins? Would they show up as differences in the Australian populations if the tested insects were not first heat stressed?

Line 304) Excellent.

Line 308) Acclimation takes time, and acclimation could be a switch. Neither this work nor Porras et al 2020 tested the effect of an initial heat shock. What would happen if you did your knockdown test, let the aphids recover, and then try again a couple of days later using the survivors? My guess is they would survive for longer.

Another difference is that Porras et al 2020 only included results from aphids that recovered after thermal shock. You did not include this step, and the aphids that would not have recovered in your experiment may not have acclimated thereby diluting the signal you were expecting.

Does the Australian isolate influence plant temperature in the cultivars that you used?

Line 308) But also differences. Different cultivars of wheat, and colony maintenance on wheat versus barley. Fertilization, soil, watering, and the quality of the growth chambers were probably different as well. Porras et al 2020 gives light intensity, and I missed that in this paper. The maintenance colony was at 20 C on barley, but the growth chamber used for experiments was at 22 C for wheat.

It is possible that I missed it, but there is no starting temperature listed in the section of the methods entitled “Virus effects on the thermal tolerance of aphids.” Additionally, there is no indication that the rate was anything other than 0.1 C per minute. If (as claimed) you both started at 20 C, then Porras would have gotten to 40 C about an hour and 20 minutes after you. I have no idea if this is enough to make a difference.

As a null hypothesis I would start with the fact that your V-R- aphids have a CTmax of almost 40, while Porras et al 2020 used aphids with a CTmax of only slightly over 35. This does not give your aphids much room to change. If you shift the box in Porras to nearly 40, the significant effect in that work would disappear. Likewise, if I drop your R-V- results to that reported in Porras et al 2020, then I might expect significant differences. My question: why are the aphids in your parent colony so thermally tolerant? If you would get an aphid from the field in the cold season (or spring) would you get the same result?

Reviewer 2 Report

Comments and Suggestions for Authors

This is extremely interesting manuscript and I am glad that I had a chance to review it. The research address the pressing scientific matters and is well set in the literature. Research is well planned, appropriate experimental and analytical methods were used. I would like to ask the authors to comment on the selection of reference gene for qPCR. The expression of single housekeeping gene might not be that stable as it has been previously believed and the changes in the relative density can be at least partially attributed to changes in expression. The discussion thoroughly explain the obtained result in he view of current literature. In my opinion this is an excellent manuscript that deserves publication in Microorganisms in current form. I could only find some minor editorial issues listed bellow:

Line 2: do not divide words in the title

Line 4 Check the guidelines for authors, marking the corresponding and first authors, use a dot after name abbreviation. Please carefully check the name spelling and affiliation to avoid later confusion.

Line 31: Please remember to change citation style to MDPI

Line 110 this -> Trojan

Line 113 Use full scientific names when they first appear in the abstract, main text or any of the table or figure caption.

Line 114: Unclear is this supposed to be a level 2 header, then please use the journal style 2.1 and for level tree headers 2.1.1.

Line 124: For plant scientific names please add the descriptor abbreviations (Medicago sativa L.)

Line 365: Please add the information about supplementary materials. Then add the citations to the references cited in the supplement, and those references to the literature. Please refer to the instructions for authors.

Round 2

Reviewer 1 Report

Comments and Suggestions for Authors

I still think that this sort of research is critical to make the scientific process work. However, there are still problems with this manuscript that make it harder to interpret than necessary. I think this is entirely a presentation issue, and can be fixed.

Your “logistical constraints” appears to be a lack of significance in the first two reps. This is p-hacking. Were there other constraints? If you could do two “blocks” at one time, why only do one the following time? (After reading the entire manuscript and response to earlier comments, I understand. However, as written this sounds like p-hacking. The presentation needs to be altered because you do not want readers to go down that road.)

Line 52, in response to reviewers) Are you asserting that the latent period is temperature independent? That does not make biological sense.

Line 142) What type of soil? Entosol, Alphisol (or clay/silt/loam)? I found this: https://apsvic.org.au/soil-types-in-victoria/ If needed you may also mention if the soil was from a disturbed site. In which case the best course would be to state the original soil type and then, to the extent possible, how it was modified.

Line 165 (original) now line 149) If this is true, then here is what you did: For the V- plants you used non-viruliferous aphids. You placed them onto the V- plants for the same length of time as V+ plants. You then discarded all of the V- plants in the V- control group. You state that “We concurrently completed the same steps for each uninoculated plant …”      This is what you say, but I will admit that people should be able to interpret what you mean.

Line 177: how do you have multiple runs within each block? A block is defined as a group of ~30 wheat plants. It does not appear that the plants are relevant here, but you still have blocks. I am confused.

Line 175) block here cannot be the same as earlier. Earlier, block was a set of 30 plants. In measuring thermotolerance, no plants were used. Alternatively, I misunderstood, and “aphids were placed in glass tubes” should read that aphids were placed in glass tubes with one of the plants from the group of 30 described earlier. Either block is not the same (and needs a better description), or there is a link between section 2.4 and 2.3 that needs to be more explicit.

Line 181) in the response to earlier comment about p-hacking. Ok, this is excellent. However, you need to change the presentation because it still sounds like p-hacking as currently written. “Following (our analysis of) these first two blocks, our results on the interaction between Rickettsiella and BYVD density of aphids remained equivocal (non-significant?) compared to our results on aphid thermotolerance. Therefore, we completed a third block …”

Line 207) It is now clear that you did not use a ZIP model. It is strange that you then cite a paper about ZIP models. Why not cite the package by citing the package documentation?

                I tried to get all the pieces to fit together. You could get a ZIP model by counting individuals as responsive versus unresponsive. This is then a format that could give ZIP type data, or could be analyzed as a probit model because it is a binary response. That is not what you did, but that is how I got confused. A title like “Generalized linear mixed models using template model builder” would not have lead me in that direction.
